

# Assessment of quantum annealing for the construction of satisfiability filters

**Marlon Azinović[1], Daniel Herr[1,2*], Ethan Brown[3], Bettina Heim[1] and Matthias Troyer[1,4]**

**1** Theoretische Physik, ETH Zurich, 8093 Zurich, Switzerland
**2** Quantum Condensed Matter Research Group, CEMS, RIKEN, Wako-shi 351-0198, Japan
**3** Mindi Technologies Ltd. 71-74 Shelton Street, Covent Garden, England
**4** Quantum Architectures and Computation Group, Microsoft Research, Redmond, WA
98052, USA

* herrd@phys.ethz.ch

## Abstract

Satisfiability filters are a new and promising type of filters for set membership testing. In order to construct satisfiability filters, it is necessary to find disparate solutions to hard random $k$-SAT problems. This paper compares simulated annealing, simulated quantum annealing and WalkSAT, an open-source SAT solver, in terms of their ability to find such solutions. The results indicate that solutions found by simulated quantum annealing are generally less disparate than solutions found by the other solvers and therefore less useful for the construction of satisfiability filters.



# 1  Introduction

Set membership testing, *i.e.* determining whether a specific element is a member of a given set in a fast and memory efficient way, is important for many applications, such as database searches. The usage of filters, such as the Bloom filter [1], can give tremendous improvements in terms of resource costs. Recently, Weaver and collaborators introduced a new type of filter, called *satisfiability filter* [2]. These are based on a specific type of Boolean satisfiability problem (SAT), namely random $k$-SAT problems, and can achieve a higher efficiency than other currently popular filters [2–4]. The creation of a satisfiability filter relies on finding solutions to SAT problems. There are many methods which can solve these SAT problems but recently with the advent of quantum annealing (QA) a fundamentally different method was introduced.

The D-Wave quantum annealer has already been used for the creation of SAT filters [5]. However, the chimera-graph of the D-Wave device does not allow arbitrary connections and thus restricts the types of problems, that can be implemented without embedding.

This paper investigates the question whether QA on an arbitrary connected and completely coherent device, can be advantageous for the creation of SAT filters.

Our simulations use an efficient implementation of simulated quantum annealing (SQA), which has been shown to be indicative of the performance of stoquastic Quantum Annealing devices [6–8]. We thus compare SQA to simulated annealing (SA) and WalkSAT (WS) [9], an open-source SAT solver.

Our results indicate that the solutions found by SQA are generally less disparate than the solutions found by the other solvers, resulting in a lower efficiency of filters constructed from them. We also find no evidence that the solutions found by SQA are particularly hard to find for the other solvers tested. Furthermore, our results show no scaling advantage of SQA over SA, regarding the computational effort required to find *one* solution with fixed probability. Thus, our simulations indicate that QA shows no improvement for the construction of SAT filters. However, it should be noted that nonlinear schedules or non-stoquastic driver Hamiltonians could still improve QA. Such research is left for further work.

In Section 2 a brief introduction to satisfiability filters and $k$-SAT problems is given. Section 3 will give a brief introduction to SA and SQA related to SAT instances and explain the measurement set-up. The results will be shown and analysed in Section 4 and concluding remarks will be provided in Section 5. Additional experimental data for $k = 3$ and $k = 5$ and results of mixing SQA-found with SA-found solutions are presented in the appendices.

# 2   Satisfiability filters

We start with a brief introduction to satisfiability filters and refer to Ref. [2] for further details.

## 2.1   The set membership problem and filters

The *set membership problem* is defined as follows: Given an element $x \in D$ and a set $Y \subseteq D$, determine whether $x \in Y$. For typical applications, $|Y|$ is very large. $Y$ could, for example, contain all words used on a specific domain and $x$ could be a key-word entered into a search engine. Such queries can be sped up using filters, which are mathematical objects created from the set $Y$. Such a filter can be queried with an element $x$ and returns one of the following answers. Either the element is definitely not in the set or the element might be in the set and further checks have to be performed to obtain a definite answer. These additional tests may be computationally more costly. Hence, a useful filter should have a low rate of indecisive answers for elements not being in the set. Furthermore, a filter should ideally require little storage.

## 2.2   Random $k$-SAT problems

The problem of deciding whether a given Boolean formula can be evaluated to true by consistently assigning the values true or false to the appearing variables is called a Boolean satisfiability problem. A *random k-SAT problem* is a Boolean satisfiability problem, which is usually given in conjunctive normal form (CNF) *i.e.* given by the logical conjunction of $m$ clauses $C_i$:

$$C_1 \wedge C_2 \wedge \cdots \wedge C_m \tag{1}$$

Each clause $C_i$ is then given by the logical disjunction of $k_i$ non-complementary literals $l_{i,j}$. Each literal is given by a Boolean variable ($x_f$) or its negation ($\bar{x}_f$), chosen from a set of $n$ variables $x_1, x_2, \ldots x_n$:

$$C_i = l_{i,1} \vee l_{i,2} \vee \cdots \vee l_{i,k_i}. \tag{2}$$

The number of literals within a clause ($k_i$ in Equation 2) is called the width of the clause. A *random k-SAT problem* is given by a Boolean formula in CNF, where each of the $m$ clauses is drawn uniformly, independently and with replacement from the set of all width $k$ clauses [10]. The problem of deciding whether a random $k$-SAT problem is satisfiable was the first problem proven to be NP-complete [11] for $k \geq 3$. Asymptotically, the satisfiability of a random $k$-SAT problem is determined with high probability by the clauses-to-variables-ratio [12], given by

$$\alpha = \frac{\text{number of clauses}}{\text{number of variables}} = \frac{m}{n}. \tag{3}$$

For every value of $k$, there exists a threshold $\alpha_k$, such that for a large number of variables, SAT instances with a clauses-to-variables-ratio $\alpha < \alpha_k$ are almost certainly solvable whereas those for which $\alpha > \alpha_k$ are almost surely not solvable. The threshold $\alpha_k$ is called satisfiability threshold and was proven to be $\alpha_k = 2^k(\ln(2) - \mathcal{O}(k))$ [13].

## 2.3   Constructing and querying a satisfiability filter

The description given here is a short summary of the explanation given in Ref. [2].

Let $Y \in D$ be the set that should be encoded by the filter and let $m$ denote the cardinality of $Y$. To construct a filter encoding set $Y$, a set of $k$ hash functions is chosen which map each element of $D$ uniformly at random into a set of $k$ distinct literals such that the literals form a clause of the form given in Equation 2, when they are combined by logical disjunction. The

set of hash functions is used to map the $m$ elements of $Y$ to $m$ clauses of width $k$. Using logical conjunctions, the resulting $m$ clauses are then combined to a random $k$-SAT problem of the form given in Equation 1. A SAT solver is then used to find $s$ different solutions to the random $k$-SAT problem constructed from set $Y$. These solutions are stored and constitute the filter.

To query the filter with an element, the same $k$ hash functions are used to map this element to a clause of width $k$. If at least one of the stored solutions does not satisfy the newly created clause, the element cannot be in the set from which the solutions where created. The hash functions, however, may map to a clause that is satisfied for all solutions by chance alone, such that a positive result does not give any further information about the set membership. One can see that more calculated solutions lead to a lower probability for an element, which is not in $Y$, to pass the filter. This constitutes a trade-off because a filter should also require little storage. Thus, to make good use of the required storage, the solutions should be as independent as possible. Here, two solutions are considered independent if the probability that a randomly generated clause is satisfied by one solution is uncorrelated with the probability that this clause is satisfied by the other one. This implies for example that the mean pairwise Hamming distance of a set of independent solutions with $n$ variables each is given by $n/2$. Finding solutions to a random $k$-SAT problem which qualify as independent is time consuming and sometimes might not be possible at all. However, this calculation is a one-time effort, which only has to be done when the filter is constructed.

To ensure that the stored solutions are independent, it is also possible to use $s$ different sets of hash functions to construct $s$ different random $k$-SAT problems out of the set $Y$. In this case, only one solution per problem has to be found. Similar to the case described before, the $s$ solutions constitute the filter. If such a filter is queried, the same $s$ sets of hash functions are used to map the element to $s$ different clauses. If the clauses are satisfied by the respective solution, the element might be in the set $Y$. The advantage of constructing a filter this way is that each stored solution stems from a different random $k$-SAT problem. Therefore, they are indeed independent of each other. The disadvantage is that more hash functions and clauses have to be evaluated every time the filter is queried, which increases the ongoing costs.

This papers compares SQA to SA and WS in terms of the ability to find multiple disparate solutions to random $k$-SAT problems. Hence, this paper focuses on the first way of constructing a satisfiability filter, which uses multiple solutions to the same random $k$-SAT problem.

## 2.4 Important quantities

We next introduce the relevant quantities needed to assess the quality of a SAT filter: the false positive rate and the filter efficiency. A more detailed explanation is given in [2].

The *false positive rate* (FPR) of a SAT filter is the probability of the filter returning an inconclusive answer (maybe) when queried with an element $x \in D$, which is not in $Y$. Therefore, the FPR is

$$p = P[F(x) = \text{maybe} \,|\, x \in D \setminus Y], \tag{4}$$

where $F(x)$ is the return of the SAT filter when queried with $x$. It can be approximated by:

$$p \approx P[F(x) = \text{maybe} \,|\, x \in D]$$
$$\approx (1 - 2^{-k})^s, \tag{5}$$

where $s$ is the number of solutions stored and $k$ the width of the clauses of the SAT instance. The first approximation is valid for $|Y| \ll |D|$ which is usually the case. The second approximation is only sensible under the assumption that the $s$ stored solutions are fairly independent. While this assumption is reasonable when the solutions stem from different SAT instances, it is less trivial when they are solutions to the same instance. However, there is experimental evidence that using $s$ different solutions to the same problem can lead to similar FPRs as for

truly independent solutions, provided their average Hamming distances are of similar magnitude. The query time decreases but the construction of the filter takes longer [2]. This paper will investigate whether QA performs better than the other solvers in finding such disparate solutions, which would justify the last approximation in Equation 5.

A SAT filter needs $n' = ns$ bits to store $s$ solutions with $n$ variables each. Thus, the *filter efficiency* of a filter encoding a set $Y$ with cardinality $|Y| = m$, having $n'$ bits of memory available and a FPR $p$, is defined [14] as

$$\mathcal{E} \equiv \frac{-\log_2 p}{n'/m} \leq 1. \tag{6}$$

In References [2, 14] Equation 5 was used to rewrite the efficiency as

$$\mathcal{E} = \frac{-\log_2(1 - 2^{-k})}{n/m} = -\log_2(1 - 2^{-k})\alpha_\chi. \tag{7}$$

Here, $\alpha_\chi$ is the clauses-to-variables-ratio of the SAT instance from which the solutions were generated. Weaver and collaborators showed that SAT filters can approach the information theoretical limit ($\mathcal{E} = 1$) by increasing the size of its clauses [2]. This also increases the complexity of the problem, but it was shown that even for relatively small values of $k = 3$ better performance compared to the constant efficiency of Bloom filters [1] ($\mathcal{E} \leq \ln(2)$) can be achieved.

Since filters are mainly used in big data contexts, their size can be quite large, depending on the size of the set $|Y|$ and the required FPR. To illustrate this, we give an example for a SAT filter encoding a set $Y$ with $|Y| = 2^{16}$ elements: The $2^{16}$ elements result in $m = 2^{16}$ clauses. If the required FPR is $p = \frac{1}{4}$ a 4-SAT filter needs to store $s \approx 22$ (independent) solutions [2]. If the targeted efficiency is given by $\mathcal{E} = 0.75$ each solution involves $n \approx 8136$ literals. The filter therefore stores $\approx 22 \cdot 8136$ bits. A 5-SAT filter achieving the same efficiency and FPR needs to store $s \approx 44$ (independent) solutions with $n \approx 4002$ literals. It should be noted that a physical implementation needs more spins as there are literals, because embedding is needed to accommodate for the restricted connectivity of physical devices. Thus, an implementation with current hardware is not yet possible, but the rapid development in this field might lead to physical implementations, soon.

Even though it is a one one-time effort to find solutions as independent as possible during the construction of sat filters, quantities such as the false positive ratio and query time heavily depend on the success of this step. Therefore, we will present different methods to obtain a solution.

## 3 Methods

This section gives a brief introduction to SA and SQA on SAT instances and a description of the conditions under which the measurements were taken, as well as some details on the implementations.

### 3.1 Simulated annealing on SAT instances

Simulated annealing (SA) was introduced by Kirkpatrick, Gelatt and Vecchi in 1983 [15]. It is a widely used optimisation method to find a configuration minimizing a given cost function. Here we describe its application to SAT instances.

Let a random $k$-SAT instance have $m$ clauses and $n$ variables. The state (configuration) of the system is defined by the Boolean value of $n$ variables $\vec{x} = (x_1, \ldots, x_n)$, where $x_i \in$

{true, false} for $1 \leq i \leq n$. In order to calculate the energy of a state (the cost of a configuration), all clauses of the SAT instance are evaluated accordingly. Each clause evaluating to true then contributes zero energy (cost), each clause evaluating to false contributes an energy (cost) of one. The energy (cost) $E(\vec{x})$ of a variable assignment $\vec{x}$ is given by the number of unsatisfied clauses

$$E(\vec{x}) = |\{C_i | C_i(\vec{x}) = \text{false}\}|, \tag{8}$$

and $|\cdot|$ denotes the cardinality of a set.

The Metropolis Monte Carlo algorithm simulates a system in thermal equilibrium at a temperature $T$. For every variable $x_i$ a proposed change $x_i \rightarrow \bar{x}_i$ is accepted with probability

$$p_{\text{acpt}}(\vec{x}_{\text{old}} \rightarrow \vec{x}_{\text{new}}) = \min\left(e^{-\beta(E(\vec{x}_{\text{new}})-E(\vec{x}_{\text{old}}))}, 1\right), \tag{9}$$

where $\beta = \frac{1}{T}$ is the inverse temperature. Hence, variable changes leading to a lower energy are always accepted, whereas the probability of accepting an increase in energy decays exponentially with increasing inverse temperature.

The annealing process consists of gradually decreasing the temperature $T$. An initial inverse temperature $\beta_0$ is linearly increased to $\beta_1$ during the process such that eventually (almost) no changes increasing the energy are accepted and the system freezes.

## 3.2 Quantum annealing on SAT instances

While SA makes use of thermal excitations to escape local minima, quantum annealing (QA) [16–20] uses quantum fluctuations to find the ground state of a system.

For the purpose of this paper, QA is performed on an Ising spin glass with $n$ distinct spins, with maximally $k$ of them coupling together. The corresponding problem Hamiltonian, $H_P$, is given by:

$$H_P = -\sum_{i_1 < \cdots < i_k} J_{i_1,\ldots,i_k} \sigma_{i_1}^z \cdots \sigma_{i_k}^z - \cdots - \sum_{i_1 < i_2} J_{i_1,i_2} \sigma_{i_1}^z \sigma_{i_2}^z - \sum_{i_1} h_{i_1} \sigma_{i_1}^z - C, \tag{10}$$

where $i_j \in \{1 \ldots n\}$ labels the $n$ spins, $J$ labels the multi-spin coupling strengths, $h_{i_j}$ denotes the local field at the position of spin $i_j$, $C$ is a constant with no physical relevance and $\sigma_{i_j}{}^r$, for $r \in \{x, y, z\}$, is the Pauli $r$-operator. In order to employ QA to find the ground state of $H_P$, a driving Hamiltonian $H_D$ is added to the problem Hamiltonian. A usual choice for $H_D$ is given by the Hamiltonian of a time dependent transverse magnetic field $\Gamma(t) \geq 0$, which induces transitions between the states $|\uparrow\rangle$ and $|\downarrow\rangle$ of a single spin:

$$H_D(t) = -\Gamma(t) \sum_{i_1} \sigma_{i_1}^x. \tag{11}$$

The total Hamiltonian is now given by

$$H_{QA}(t) = H_P + H_D(t). \tag{12}$$

To perform QA on a SAT instance, each variable is identified with a spin. The couplings for $H_P$, are chosen so that its energy is equal to the number of unsatisfied clauses for that assignment. The ground state of $H_P$ will then provide a solution to the SAT instance, and the energy landscape of $H_P$ is the same as specified in Equation 8 for the case of SA.

At the beginning of the annealing schedule $\Gamma(0) > |J_{\ldots}|, |h_{i_j}|$, so that $H_{QA}$ is dominated by $H_D$ and the system will be in the easily attainable ground state of $H_D$ with all spins pointing along $x$-direction, that is each variable is in a superposition between true and false. Then, the transverse magnetic field is slowly decreased, reducing the tunneling rates while the system explores the configuration space. For our simulations we decreased the magnetic field linearly.

At the end of the annealing schedule $\Gamma(t_{\text{end}}) = 0$ and $H_{QA}(t_{\text{end}}) = H_P$, such that the system freezes in a configuration that provides a guess for a solution of the SAT instance.

Even though there is evidence that QA only finds few of the ground states for problems with high degeneracy (see Ref. [21] for a theoretical prediction based on quantum Monte Carlo (2009) and Ref. [22] for a recent experimental verification on a D-Wave 2X quantum annealing machine), it might still be that the found solutions are particularly useful to construct satisfiability filters. In this paper idealized QA is considered that allows for arbitrary couplings between two or more spins without the need for embedding.

**Simulated quantum annealing**

QA for spin glass models in a transverse magnetic field can be mimicked on a classical computer using *Quantum Monte Carlo* (QMC) simulations as described in [23–25], which we call simulated quantum annealing. A Suzuki-Trotter decomposition [25] is used to map a $d$-dimensional quantum spin glass model in a transverse field $\Gamma$ to a $(d + 1)$-dimensional classical model with an additional imaginary time dimension. Along the imaginary time direction, the system is discretized into $M$ time slices (TS), resulting in imaginary time-steps of width $\Delta_\tau = \beta/M$ and a coupling of strength $-\frac{1}{2\beta} \log \tanh \Gamma \Delta_\tau$ between the TS. During the annealing process, cluster updates in imaginary time direction are used.

Even though a discretization with $\Delta_\tau = 1$ might produce better optimization results, in order to simulate the real quantum behavior, computations should be performed in the continuous time limit $\Delta_\tau \to 0$ [26]. It is also possible to use an algorithm directly working with an infinite number of time slices, as in [27], providing the same results as QMC using sufficiently small time-steps. As Ref. [26] shows, previous expectations of quantum speedup for QA in two-dimensional spin glasses [23, 24] were due to performing SQA in the non-physical limit, which elegantly explains why such a speed-up could not be observed in experiments [28, 29]. For the purpose of this paper, SQA is performed using QMC close enough to the continuous time limit to guarantee similar results to a continuous time algorithm and a physical quantum annealer.

Recent work [6] shows a similar scaling of characteristic tunneling rates for SQA and unitary evolution, and suggests that SQA can indeed to a certain extent predict the performance of a physical quantum annealer. Additionally D-wave quantum annealers where empirically shown to scale similarly to SQA for random spin glass problems [7] and a numerical partitioning problem [8].

### 3.3 Annealing parameters of SA and SQA

For each problem specification and each given number of MCS, the annealing parameters for SA ($\beta_0$ and $\beta_1$) were optimized in the sense that they maximize the average number of different results for a set of instances when running multiple times on each instance. The number of MCS was chosen such that an increase would not lead to a significantly larger number of different solutions found by any of the tested solvers. This value was chosen the same for SA and SQA. The SQA parameters ($\Gamma_0$, $\Gamma_1$, $\beta_0 = \beta_1$ and $M$) were optimized the same way, ensuring that $T_0 \times M$ is big enough to be close to the physical the continuous time limit. Here, one MCS corresponds to one attempted update per physical spin (*i.e.* a certain site on all replicas). Between MC steps the transverse field was decreased linearly. The temperature was held constant during the annealing process and open boundary conditions were used, since this improves convergence [6]. When the scaling in problem size is considered, the annealing parameters were optimized in the sense that they minimize the computational effort needed for finding a solution with a 99% chance.

For a given $k$ and a given clauses-to-variables-ratio, any quantity, for example how many runs are needed to find a given number of solutions, might still vary between randomly generated instances. Due to limitations in computing resources, it is however not possible to perform all calculations for a huge number of instances. Therefore, this paper considered 20 randomly chosen instances for each specification. The relevant quantities were individually calculated for each instance and the estimate for the problem class was calculated by taking the average over the instances. The errorbars indicate the standard error of the mean.

### 3.4 WalkSAT

WalkSAT (WS) [9] is an efficient open source SAT solver, also used in [2]. In this paper version 51 from [30] was used. It was run with the flags

```
-printonlysol=TRUE, -out output_filename, -seed SEED
```

Here, in order to obtain multiple solutions, SEED was a different positive natural number for every run on the same SAT instance.

## 4 Results and discussion

In order to construct a filter from solutions to a single $k$-SAT instance, it is necessary to find multiple and disparate solutions to that instance (see Section 2.2). The number of existing solutions and how different they are depends on how close to the satisfiability threshold the SAT instance is. Hence, we first fix the efficiency to a desired value and investigate both aspects.

For different values of $k$, 20 SAT instances with 50 variables were randomly generated. Then, all solvers were run multiple times on each instance, after optimizing both SQA and SA parameters. Since this paper mainly investigates what kind of solutions are found by each solver, the runtime is not considered here.

### 4.1 Number of different solutions found

We first investigate how many different solutions can be found within a given number of repetitions of various solvers. Figure 1 shows the average number of different solutions found when running each solver 2000 times with optimized annealing parameters on a randomly generated 4-SAT instance with $n = 50$ variables and $m = 403$ clauses. If independent solutions could be assumed, this clauses-to-variables-ratio would lead to an efficiency of $\mathscr{E} \approx 0.75$. The number of found solutions is averaged over 20 randomly generated instances, where the value for each instance is reweighted by dividing it through the number of solutions found by SA within 2000 runs. From Fig. 1 it can be seen that the effort needed for a new solution increases with the number of already found solutions. This assumes that no further techniques such as blocking clauses (see Section 4.3) are employed to prevent repeated solutions. The number of different solutions found by SA is bigger than the number of different solutions found by SQA. Using WS naively, by just providing different random seeds, it finds the smallest number of different solutions.

The same measurements were performed for 3-SAT and 5-SAT instances at a clauses-to-variables-ratio which would lead to the same efficiency of $\mathscr{E} \approx 0.75$, if independent solutions could be assumed (see Appendix B and Appendix C). In the $k = 3$ case, the ranking among the solvers is similar. For $k = 5$, SQA finds the smallest number of different solutions, especially close to the satisfiability threshold.

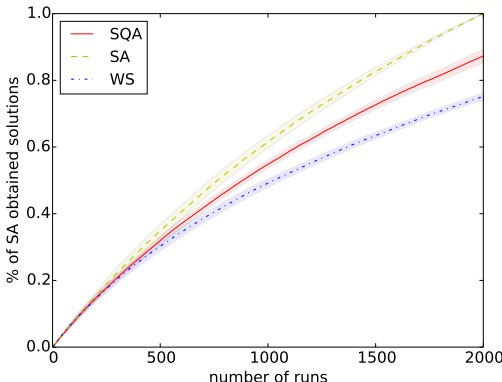

Figure 1: Average number of different solutions obtained, when running each solver 2000 times on each instance. The average is taken over 20 randomly generated 4-SAT instances with $n = 50$ variables and a clauses-to-variables-ratio of $\alpha_{\chi_4} \approx 8.06$. The number of different solutions obtained is normalized by the maximum number of different solutions found by SA within 2000 runs (about 1300).

## 4.2 Quality of the solutions

The next question to answer is whether the solutions found by the different solvers are of the same *quality* for SAT filter construction. The quality of found solutions is determined by the FPR of a filter constructed from these solutions, which we show in Fig. 2. In Fig. 2a) we choose the solutions randomly. Fig. 2b) shows a subset of solutions with particularly large Hamming distance between each other[1]. The results show that the solutions found by SQA lead to a *higher* FPR than the solutions found by SA and WS. If more than ten solutions are chosen, all solvers provide a significantly worse FPR than truly independent solutions.

The corresponding filter efficiencies are shown in Fig. 3. Consistent with the higher FPR, the efficiency of a filter constructed using SQA is lower than the efficiency of a filter constructed using SA or WS. Fig. 3 also shows that only for a small number of solutions, the efficiency of the constructed filter is comparable to the efficiency resulting from independent solutions, and only if they are specifically selected. The maximum number of solutions that still leads to an efficiency comparable to truly independent solutions, is about five for SQA and between seven and ten for SA and WS. This result indicates that combining the two techniques suggested in [2] might be useful: Instead of using either multiple sets of hash-functions and one solution per instance or one hash-function and many solutions of the resulting instance, it might be preferred to take multiple sets of hash functions and still find multiple solutions per instance.

The reason for the differences in the FPRs can be further investigated by looking at the average or maximum Hamming distance between solutions. The average (Fig. 4a)) and maximum (Fig. 4b)) Hamming distance, taken pairwise between different solutions found by the different solvers is shown in Fig. 4, averaged over 20 instances. The result shows that the subset of solutions found by SQA have a much higher overlap than the subset of solutions found by SA and WS, resulting in a higher FPR of the SQA solutions. Without imposing further requirements, the approximation in Equation 5 is worse for SQA than for the other solvers. We also did not find evidence that SQA finds solutions that feature a particularly large Hamming distance to the set of solutions found by the other solvers.

---

[1]In order to obtain this subset, a random solution was chosen initially. Afterwards, solution by solution, the solution with the highest average pairwise Hamming distance to the already chosen solutions was added to the set.

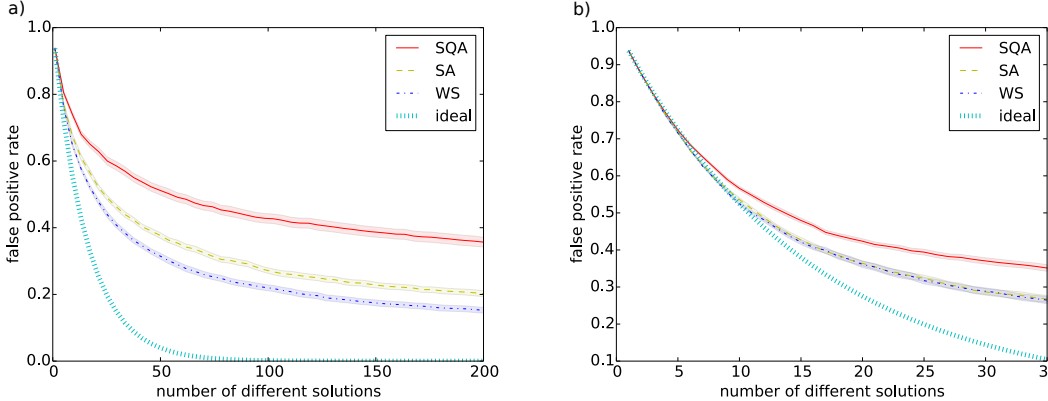

Figure 2: Average FPR of a filter constructed from solutions found by each solver within 2000 runs. The average is taken over 20 randomly generated 4-SAT instances with $n = 50$ variables and a clauses-to-variables-ratio of $\alpha_{\chi_4} \approx 8.06$. a) The solutions used to construct the filter are chosen randomly from the found solutions. b) The solutions used to construct the filter are chosen so that they maximize the average pairwise Hamming distance between the chosen solutions. The dotted cyan line shows the theoretical FPR for a filter constructed from truly independent solutions.

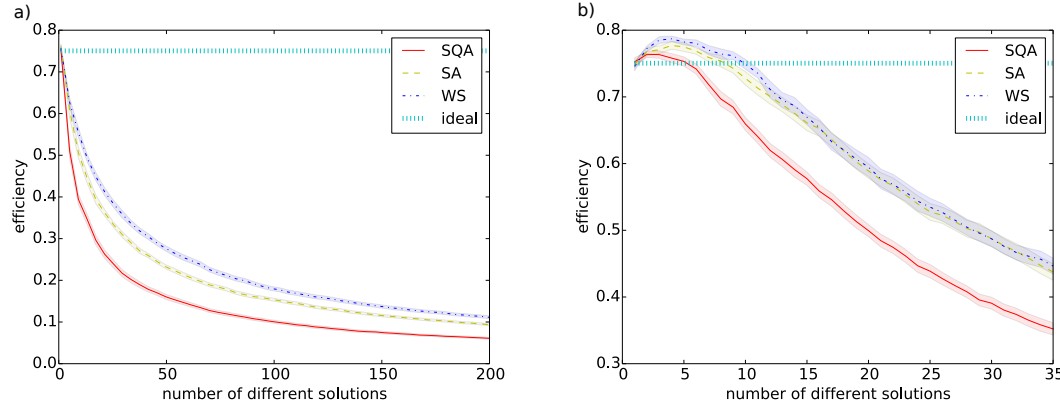

Figure 3: Average efficiency of a filter constructed from solutions found by each solver within 2000 runs. The average is taken over 20 randomly generated 4-SAT instances with $n = 50$ variables and a clauses-to-variables-ratio of $\alpha_{\chi_4} \approx 8.06$. a) The solutions used to construct the filter are chosen randomly from the found solutions. b) The solutions used to construct the filter are chosen so that they maximize the average pairwise Hamming distance between the chosen solutions. The dotted cyan line shows the theoretical FPR for a filter constructed from truly independent solutions.

If SQA could easily find solutions that are hard to find for the other solvers, it could be used to contribute complementary solutions. To study whether SQA finds solutions that are hard to find for other solvers, Fig. 5 investigates how easily WS and SA can obtain solutions found by SQA. As can be seen in Fig. 5, we find no evidence that the solutions which are easily found by SQA, are particularly unlikely to be found by SA (Fig. 5b)) and WS (Fig. 5a)). Here we provide an average over a percentile of solutions. For future studies, it would be interesting to compare the probability with which each solver finds each of the existing solutions. This

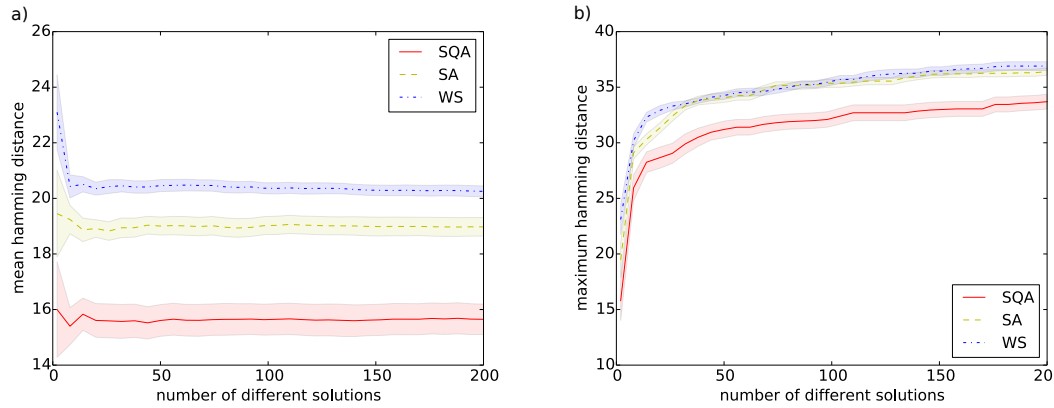

Figure 4: Pairwise Hamming distance between different solutions randomly picked from the solutions obtained by each solver within 2000 runs. The Hamming distance was calculated for up to 200 different solutions found for each of 20 randomly generated 4-SAT instances with $n = 50$ variables and a clauses-to-variables-ratio of $\alpha_{\chi_4} \approx 8.06$. An average was taken over the instances. a) Shows the average pairwise Hamming distance between the picked solutions. b) Shows the maximum pairwise Hamming distance between the picked solutions.

would require far more runs and is out of the scope of our study.

For all data presented so far, analogous plots for $k = 3$ and $k = 5$ are provided in Appendices B and C. It can be seen that a choice of parameters, which results in the same efficiency, leads to the same rankings and conclusions. Further evidence that mixing SQA-found solutions with solutions found by other solvers does not improve the resulting filter is provided in Appendix A.

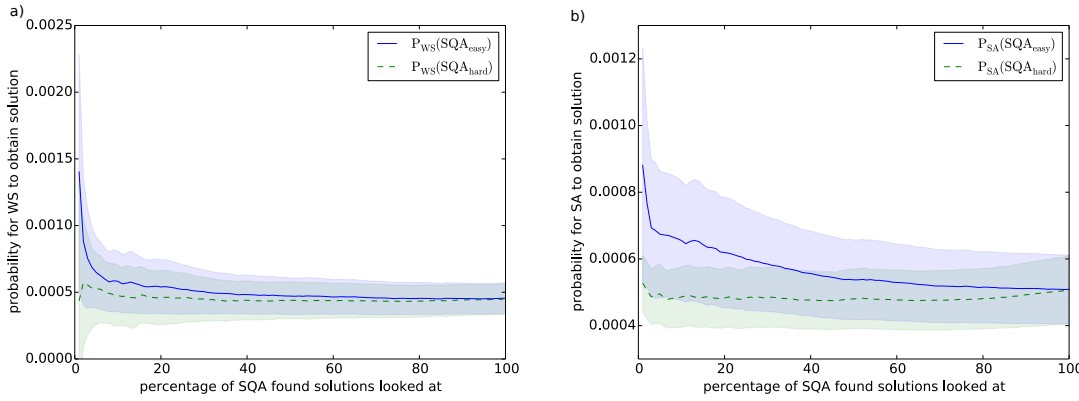

Figure 5: Probability for a) WS and b) SA to find the SQA-hardest and SQA-easiest solutions. The x-axis indicates the percentage of easiest and hardest SQA-found solutions, running SQA 2000 times on 20 randomly generated 4-SAT instances with $n = 50$ variables and a clauses-to-variables-ratio of $\alpha_{\chi_4} \approx 8.06$. An average was taken over the instances.

## 4.3 Scaling with problem size

Next we compare the solvers abilities to find *one* solution to a given SAT problem. This is relevant if the filter is constructed by finding one solution to several different instances and

might also be indicative of the solvers performance when blocking clauses [31] are used. A blocking clause is a clause which evaluates to *true* for all variable assignments except for some specific assignments, which are not desired. Adding a blocking clause to a random *k*-SAT problem removes all solutions with the undesired assignments of variables from the set of solutions. Blocking clauses can give a positive contribution to the energy if the configuration has already been found. Therefore, any solution found to the new problem, is necessarily different from the solutions previously obtained. It is also possible to add clauses which not only block a certain solution, but certain variable assignments. By blocking certain variable assignments it can be ensured that any solution to the new problem has a certain Hamming distance to the solutions found previously. However, these procedures make it less likely for a solver to still be able to find solutions.

The SAT competitions [32,33] compare many highly optimized classical solvers in a variety of related tasks. In order to evaluate if SQA has the potential to speed up this task, it will be compared to SA only, since the natural similarity between SA and SQA makes it easy to compare their performance.

Figure 6 shows the computational effort needed to find *one* solution to a 4-SAT instance with a probability of 99% for different problem sizes at $\alpha = 8.06$. For every system size, 20 instances were randomly generated. Each solver is run 60 times on each instance with optimized parameters and performance was averaged over the 20 instances. Since we are interested in scaling rather than absolute performance we measure time in units of Monte Carlo steps, although their complexity is different for SQA, SA, and physical QA.

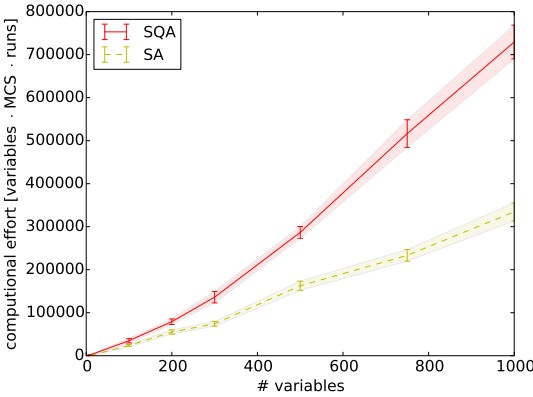

Figure 6: Average computational effort needed to find a solution to a random 4-SAT problem with 99%. The average is taken over 20 randomly generated 4-SAT instances with a clauses-to-variables-ratio of $\alpha_{\chi_4} \approx 8.06$.

Consistent with the findings of Ref. [34–37], Fig. 6 provides no evidence that SQA requires a lower computational effort to find one solution for all tested problem sizes. There is no indication that SQA scales better than SA for random SAT instances, either. It can therefore be concluded that SQA, for the tested instances and using a linear schedule with a stoquastic driver Hamiltonian, is not superior to SA, also when it comes to finding *one* solution to a SAT instance. The study of the computational effort required to find one solution for higher values of *k* and higher clauses-to-variables-ratios, is left for future research.

# 5 Conclusion

The focus of this paper has been the assessment of SQA and QA as a SAT solver for the construction of satisfiability filters. We found that SQA finds a smaller number of different solutions than SA and that the found solutions have a smaller Hamming distance, which leads to a higher FPR and lower efficiency. Thus, SQA performs worse than other solvers on any metric that was investigated. The solutions found by SQA are also not particularly difficult to find for other solvers. Thus, our simulations show no indication that simulated quantum annealing is better than simulated annealing for the construction of SAT filters.

The simulations in this paper are best-case scenarios for analog QA devices in the sense that even though we restricted our simulations to a simple enough driver Hamiltonian, we still assumed hardware with arbitrary connectivity and multi-spin couplings along the z-direction. If the problem is mapped to a specific hardware graph and few-qubit couplers, as it is necessary for physical QA devices, the performance of QA will further decrease. A D-Wave Two device has recently been used to construct variants of SAT filters [5], by solving variants of SAT problems, e.g. *Not-all-equal 3-SAT*, which can be realized with two-qubit couplings.

However, we did not investigate nonlinear annealing schedules, which might improve the efficiency of QA and SQA. Another way to improve QA and SQA is to find a method which enlarges the subset of found solutions. As theoretically discussed in Ref. [21] and indicated by recent experimental results [22] using a D-Wave 2X quantum annealing machine, QA needs more complex (non-stoquastic) driving Hamiltonians in order to find degenerate ground states with equal probability. Furthermore, Ref. [21] predicts that QA can improve, if quantum transitions between all degenerate ground states are possible. This can be done by modifying the driver Hamiltonian in Equation 11, so that it induces quantum transitions between all states:

$$H_D^{\text{new}}(t) = -\Gamma(t)\Big(\sum_{i_1} \sigma_{i_1}^x + \sum_{<i_1,i_2>} \sigma_{i_1}^x \sigma_{i_2}^x + \sum_{<i_1,i_2,i_3>} \sigma_{i_1}^x \sigma_{i_2}^x \sigma_{i_3}^x + \dots\Big).$$

Unfortunately, such terms are very difficult to implement both in simulations and in experimental devices.

The previously mentioned nonlinear schedules and non-stoquastic driver Hamiltonians could improve the performance considerably and should be investigated in further research. This research would not only be important for analog devices such as the D-Wave devices but would also be fairly indicative of the performance of QA run on a digital quantum computer, where implementation of more complex driving Hamiltonians should be easier.

Generally, it is more difficult to achieve low FPRs using only one set of hash-functions than when using multiple sets of hash-functions. We showed that all solvers are able to find a certain number of solutions, which are as good as independent. Therefore, for the general use of SAT filters, it might be recommendable to combine these approaches and use multiple sets of hash-functions and multiple solutions per instance in order to construct the filter.

# Acknowledgments

We thank Ilia Zintchenko for providing the simulated annealing code as well as the translation of a *k*-SAT problem to an Ising spin glass problem. We thank Guglielmo Mazzola, Giuseppe Carleo, José Luis Hablützel Aceijas and Andreas Elsener for helpful discussions. We thank Ilia Zintchenko and Alex Kosenkov for providing and supporting the *whiplash* framework to run all simulations presented in this paper. MT acknowledges hospitality of the Aspen Center for Physics, supported by NSF grant PHY-1066293. This paper is based upon work supported in part by ODNI, IARPA via MIT Lincoln Laboratory Air Force Contract No. FA8721-05-C-

0002. The views and conclusions contained herein are those of the authors and should not be interpreted as necessarily representing the official policies or endorsements, either expressed or implied, of ODNI, IARPA, or the U.S. Government. The U.S. Government is authorized to reproduce and distribute reprints for Governmental purpose not-withstanding any copyright annotation thereon.

## A  Mixing SQA and SA solutions

| SQA, SQA | SA, SA | WS, WS | SQA, SA | SQA, WS | SA, WS |
|---|---|---|---|---|---|
| 15.6 | 18.9 | 20.2 | 17.9 | 19.1 | 19.7 |
| ±1.4 | ±0.6 | ±0.2 | ±1.1 | ±0.3 | ±0.4 |

Table 1:  This table shows the average Hamming distance between two solutions, found by different solvers, of a randomly generated 4-SAT. The value provided is the average taken over 20 randomly generated instances with $n = 50$ and $\alpha_{\chi_4} \approx 8.06$. The solutions were randomly chosen among the solutions found within 800 runs by the respective solvers. Truly independent solutions would have an average Hamming distance of 25.

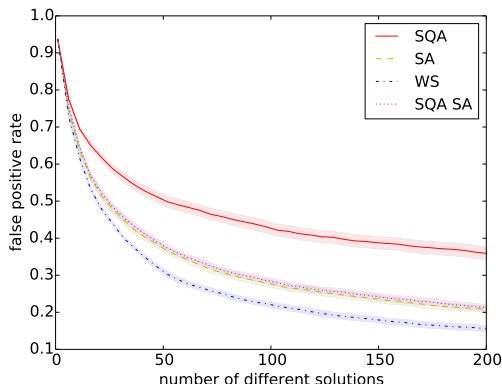

Figure 7:  Average FPR of a filter constructed from solutions randomly chosen from the set of solutions found by each solver within 800 runs. The average is taken over 20 randomly generated 4-SAT instances with $n = 50$ variables and a clauses-to-variables-ratio of $\alpha_{\chi_4} \approx 8.06$. For SQA SA, 10% of the solutions were chosen from the solutions found by SQA and 90% were chosen from the solutions found by SA.

Calculating the average Hamming distance between a solution found by SQA and a solution by one of the other solvers we find the results shown in Tab. 1 for $k = 4$, $n = 50$, $\alpha_{\chi_4} \approx 8.06$. This confirms the previously obtained results that the solutions found by SQA feature the lowest average Hamming distance among themselves while the solutions found by WS have the highest diversity. Furthermore, Tab. 1 shows that SQA-found solutions do not have a particularly large average Hamming distance to SA or WS solutions. This result gives further evidence that mixing SQA-found solutions with solutions found by the other solvers may not lead to an improved FPR. Indeed, constructing such a filter, choosing a certain percentage of SQA-found solutions and mixing them with SA-found or WS-found solutions does not lead to a lower FPR. In Fig. 7, an example for the achieved FPR of a filter constructed from 10%

a)

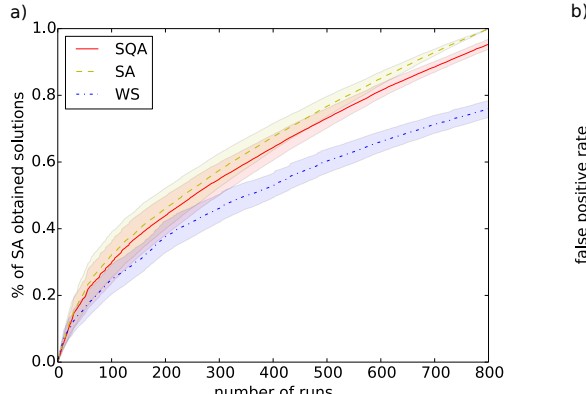

b)

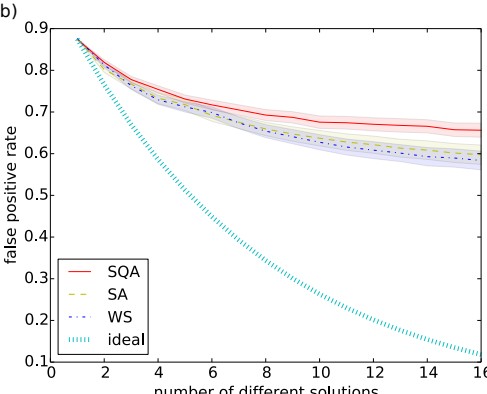

Figure 8:   The plots are averaged over 20 randomly generated 3-SAT instances with $n = 50$ variables and a clauses-to-variables-ratio of $\alpha_{\chi_3} \approx 3.9$. In a) a normalized average of the different solutions is shown and in b) the FPR is shown.

SQA-found and 90% SA-found solutions is given. An increase in the percentage of SA solution leads to the SQA-SA curve being closer to SA and a reduction leads to the SQA-SA curve being closer to SQA. No percentage of SQA-found solutions leads to a lower FPR than taking SA-found solutions only.

## B   Experimental results for $k = 3$

For $k = 3$ a clauses-to-variables-ratio of $\alpha_{\chi_3} = 3.9$ has to be chosen to obtain an efficiency of 0.75 for independent solutions. In order to obtain this efficiency for $k = 3$, one has to go closer to the threshold than for $k = 4$. Thus, the 3-SAT instances generally have less solutions than the 4-SAT instances. This makes it harder or impossible to reach a desirable FPR. When generating 20 instances at random, some of them only feature a small number of possible solution. This leads to a larger variance between different instances for the quantities looked at. For the 20 randomly generated instances, one of them only had 16 different solutions, many less than 100. This is the reason why in this section, quantities like the FPR or the Hamming distance can only be conclusively displayed for at most 16 different solutions.

Similar to Fig. 1, Fig. 8 a) shows the number of different results obtained. Here, the number of solutions found for each instance was normalized by dividing it through the number of solutions found by SA within 800 runs, which reduces the variance between the instances. The parameters were optimized to find the most different solutions. The figure shows the same ranking among the solvers as for $k = 4$.

Figure 8 b) shows the FPR. Since the largest number of solutions found by all solvers for all instances is 16, only up to 16 solutions were chosen. Comparing this figure to Fig. 2, one can observe that in the $k = 3$ case the difference between the obtained FPR and the FPR for truly independent solutions is bigger than for $k = 4$. This can be attributed to the fact that the instances are closer to the satisfiability threshold, leading to a smaller number of solutions and a higher correlation among them. When constructing SAT filters, one should therefore keep in mind that in order to achieve a given efficiency, one should choose $k$ large enough. The larger $k$ is, the more independent solutions exist. Here also, the ranking among the solvers is the same for $k = 3$ and $k = 4$.

The resulting efficiency is given in Fig. 9. Here, the ranking among the solvers is the same

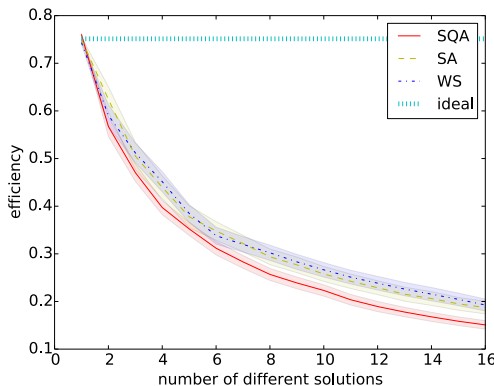

Figure 9: Average efficiency of a 3-SAT filter with a clauses-to-variables-ratio of $\alpha_{\chi_3} \approx 3.9$. The solutions used to construct the filter were chosen randomly from the found solutions.

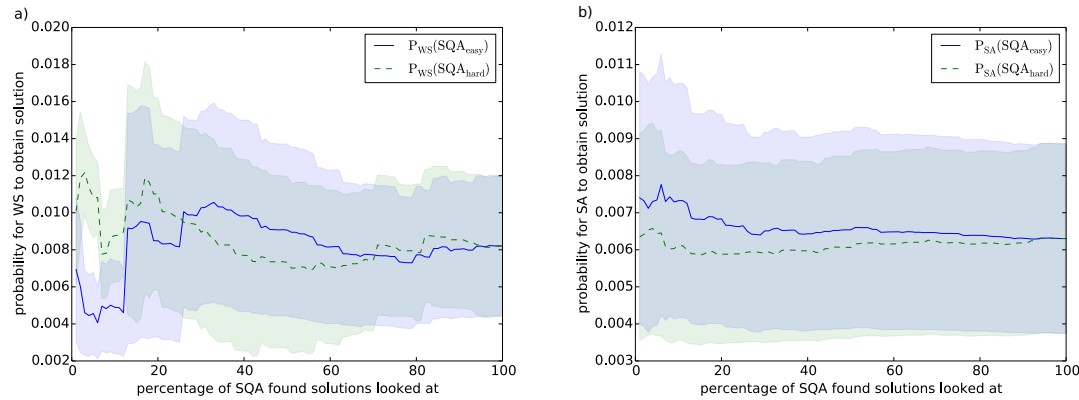

Figure 10: Probability for a) WS and b) SA to find the SQA-hardest and SQA-easiest solutions. The x-axis indicates the percentage of easiest and hardest SQA-found solutions, running SQA 800 times on 20 randomly generated 3-SAT instances with $n = 50$ variables and a clauses-to-variables-ratio of $\alpha_{\chi_3} \approx 3.9$. An average was taken over the instances.

as in the $k = 4$ case. Since the Hamming distance and the FPR are closely related, one can thus conclude that the qualitative behavior for the Hamming distance is the same between $k = 4$ and $k = 3$. Similarly to $k = 4$, for $k = 3$, we find no evidence that the solutions found by SQA are particularly hard or easy to find for the other solvers. Fig. 10 shows how difficult it is for WS (Fig. 10a)) and SA (Fig. 10b)) to find the SQA-easiest or SQA-hardest solutions.

## C Experimental results for $k = 5$

The same simulations were also carried out for $k = 5$ and $\alpha_{\chi_5} = 16.36$, which would lead to an efficiency of 0.75 if independent solutions could be assumed. Further simulations were performed with $\alpha_{\chi_5} \approx 19.6$, which would lead to an efficiency of 0.9. In order to obtain a given efficiency for $k = 5$, one can generate the instances further away from the satisfiability threshold than for $k = 4$ or $k = 3$. The case $\alpha_{\chi_5} \approx 19.6$ is close to the satisfiability threshold $\alpha_5 \approx 21.11$ of 5-SAT [38]. Hence, many of the randomly generated instances only feature a

a)

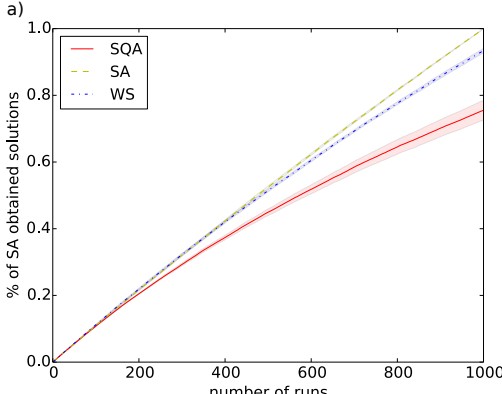

b)

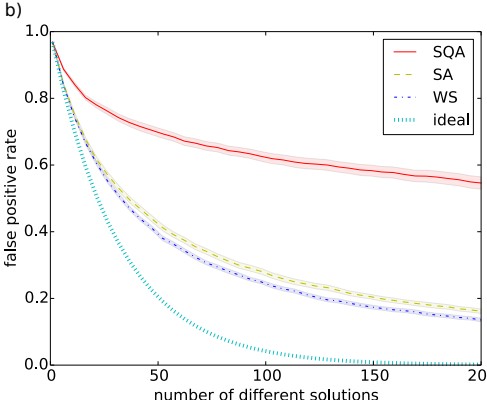

Figure 11: The plots are averaged over 20 randomly generated 5-SAT instances with $n = 50$ variables and a clauses-to-variables-ratio of $\alpha_{\chi_5} \approx 16.36$. In a) a normalized average of different solutions is shown and in b) the FPR is shown.

small number of existing solutions. Naturally, the assumption of independence is particularly bad as can be seen below.

For the case where $\alpha_{\chi_5} = 16.36$, the instances are far enough from the threshold for some solutions to exist, which leads to a similar FPR as truly independent solutions. The results for that case are shown in Appendix C.1. Generally, the rankings among the solvers stay similar for $k = 3$, $k = 4$ and $k = 5$. The data provides evidence that an advantage of SQA over SA for higher values of $k$ is not to be expected.

## C.1 Clauses-to-variable ratio 16.36

Figure 11 a) shows the number of different results obtained when running each solver 1000 times on each of 20 randomly generated 5-SAT instances. Again, SA found the most different solutions. Here, unlike for $k = 4$, SQA found the fewest.

Figure 11 b) shows the FPR of a filter constructed from randomly selected solutions from the solutions found within 1000 runs. Comparing Fig. 11 b) to Fig. 2 and Fig. 8 b), one can observe that in the $k = 5$ case the obtained FPR and the FPR for truly independent solutions are similar up to a larger number of solutions than for $k = 4$ and $k = 3$. This can be attributed to the fact that the instances are further away from the satisfiability threshold, leading to a larger number of solutions and a lower correlation among them. Here, the ranking among the solvers is the same for $k = 3$ and $k = 4$.

The resulting efficiency is given in Fig. 12. Again, the ranking among the solvers is the same as in the $k = 4$ case. Similarly to $k = 4$ and $k = 3$, for $k = 5$, we find no evidence that the results found by SQA are particularly hard to find for the other solvers, as can be seen in Fig. 13.

## C.2 Clauses-to-variable ratio 19.6

For the simulations performed at this clauses-to-variables ratio, the number of MCS was taken the same as for the lower ratio, without confirming if an increase in MCS could improve the performance. The thorough study of higher clauses-to-variables ratios and higher values of k, as well as an analysis of the computational effort required to find one solution, is left for future research.

The instances created this close to the threshold have only a small number of solutions.

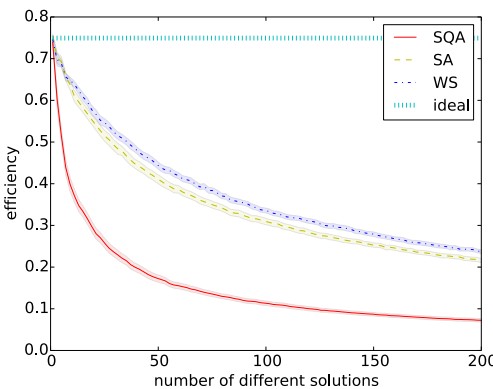

Figure 12: Average efficiency of a 5-SAT filter with a clauses-to-variables-ratio of $\alpha_{\chi_5} \approx 16.36$. The solutions used to construct the filter were chosen randomly from the found solutions.

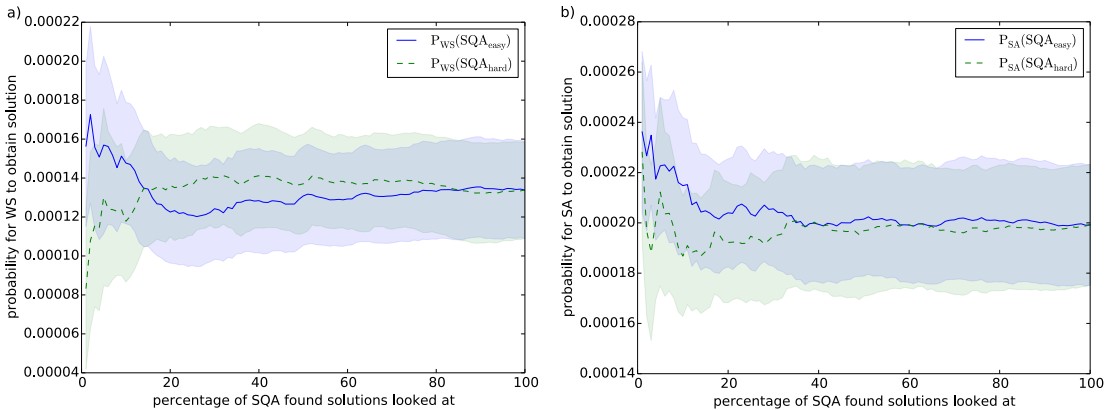

Figure 13: Probability for a) WS and b) SA to find the SQA-hardest and SQA-easiest solutions. The x-axis indicates the percentage of easiest and hardest SQA-found solutions, running SQA 1000 times on 20 randomly generated 5-SAT instances with $n = 50$ variables and a clauses-to-variables-ratio of $\alpha_{\chi_5} \approx 16.36$. An average was taken over the instances.

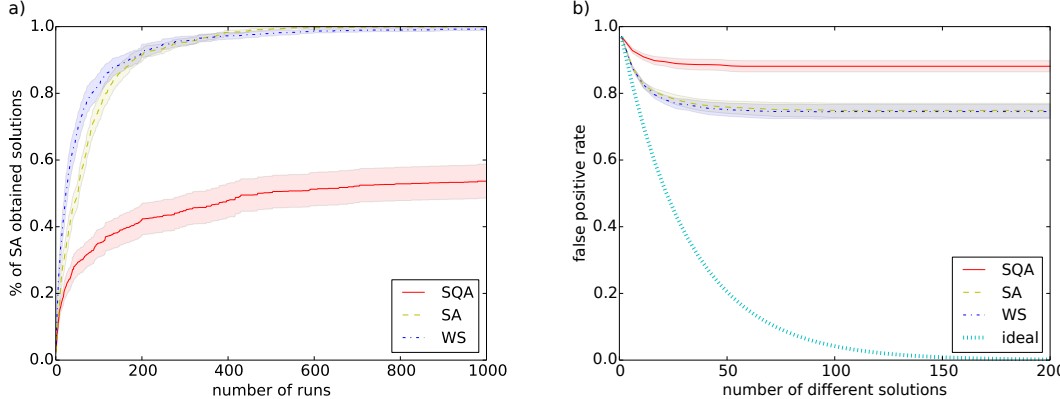

Figure 14: The plots are averaged over 20 randomly generated 5-SAT instances with $n = 48$ variables and a clauses-to-variables-ratio of $\alpha_{\chi_5} \approx 19.6$. In a) a normalized average of the different solutions is shown and in b) the FPR is shown.

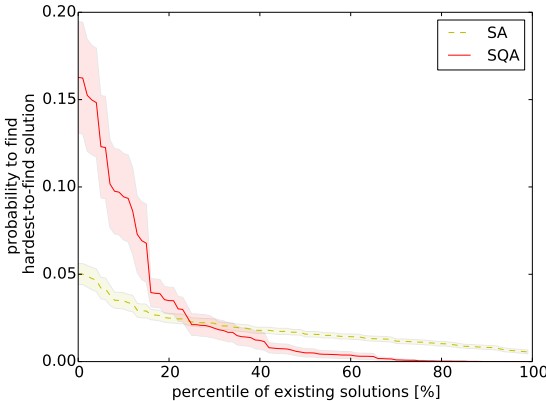

Figure 15: Average probability to find the hardest-to-find solution among the most often found percentile of all existing solutions. The average is taken over 20 randomly generated 5-SAT instances with a clauses-to-variables-ratio of $\alpha_{\chi_5} = 19.6$.

For 20 randomly generated instances, the mean number of existing solutions was 40, two instances had less than ten. For all instances SQA-found the least different solutions within 1000 runs.

Figure 14 a) shows the normalized number of different solutions obtained when running each solver 1000 times on each of 20 randomly generated 5-SAT instances. The results again show that SQA finds a significantly fewer different solutions than the other two solvers. For 19 out of the 20 instances considered here, SA found all existing solutions. Therefore Fig. 14 a) can also be understood as the percentage of existing solutions found (y-axis) within a given number of runs (x-axis). This result confirms again that SQA tends to find only a fraction of degenerate ground states [21, 22].

The fact that SQA finds only a fraction of degenerate ground states can also be observed by looking at the probability with which solutions are found. Fig. 15 shows the average probability to find the hardest-to-find solution among the most often found percentile of all existing solutions. As Fig. 15 shows, the probability to find a given solution is more uniformly distributed for SA than for SQA. This again confirms the findings in Ref. [21] and Ref. [22].

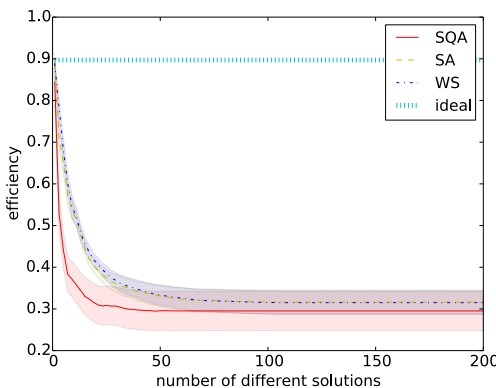

Figure 16: Average efficiency of a 5-SAT filter with a clauses-to-variables-ratio of $\alpha_{\chi_5} \approx 19.6$. The solutions used to construct the filter were chosen randomly from the found solutions.

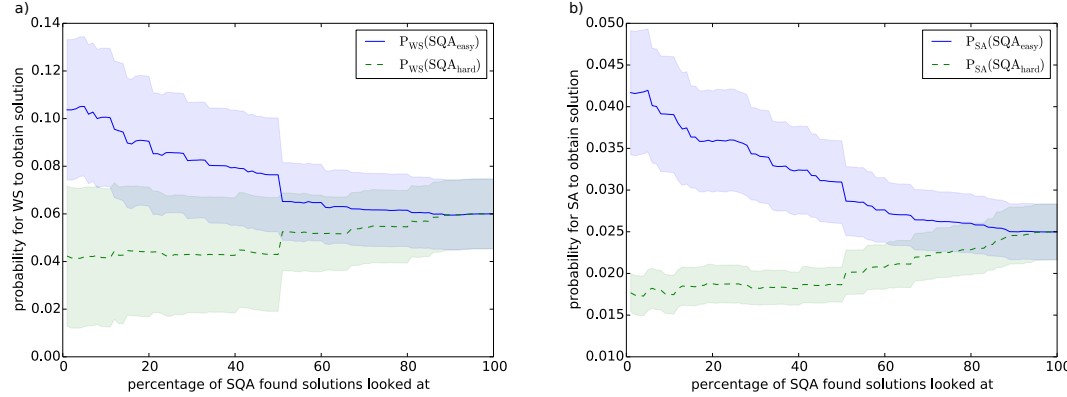

Figure 17: Probability for a) WS and b) SA to find the SQA-hardest and SQA-easiest solutions. The x-axis indicates the percentage of easiest and hardest SQA found solutions, running SQA 1000 times on 20 randomly generated 5-SAT instances with $n = 48$ variables and a clauses-to-variables-ratio of $\alpha_{\chi_5} \approx 19.6$. One instance, for which SQA did not find any solution, was taken out of the statistic. An average was taken over the instances.

Figure 14 b) shows the FPR of a filter constructed from the solutions found within 1000 runs. These solutions were again selected randomly. If a given number of different solutions was not found by a solver for an instance, the respective FPR was held constant. The same was done for the efficiency (Fig. 16). Comparing Fig. 14 b) to Fig. 11 b), one can observe that for $\alpha_{\chi_5} \approx 19.6$ the difference between the obtained FPR and the FPR for truly independent solutions is much larger than for $\alpha_{\chi_5} \approx 16.36$. This can again be attributed to the fact that the instances are closer to the satisfiability threshold, leading to a lower number of solutions and a higher correlation among them. The ranking among the solvers is similar for both clauses-to-variables-ratios as well as for all tested values of $k$.

The resulting efficiency is given in Fig. 16 and also features the same ranking among the solvers. Similarly to $k = 4$ and $k = 3$, for $k = 5$, there is no indication that SQA is able to find solutions which are particularly hard to find for the other solvers, as can be seen in Fig. 17.

### C.3   Increasing the targeted efficiency

The results shown in this paper also indicate that, as long as no further techniques like blocking clauses are employed, the approach to look for many solutions of one instance is generally not useful when targeting a high efficiency. In order to obtain a high efficiency the instances have to be generated too close to the satisfiability threshold for many sufficiently disparate solutions to be found. A FPR as predicted by assuming independent solutions is hence not achievable. The worse performance of SQA in terms of finding disparate solutions becomes more apparent for these instances, which only have a small number of solutions.

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
