# Peer review of "Assessment of Quantum Annealing for the Construction of Satisfiability Filters"

_SciPost Physics, doi:SciPost Phys. 2, 013 (2017)_

## Round 3 · Referee Report · Anonymous · 2016-11-21

Strengths
I have a strong background in Satisfiability, but not much knowledge of Quantum Physics. So, I decline to provide comments on Section III.
1. The connection between these new filters and quantum physics is really cool.
2. The paper explores in depth the original SAT filter type (clauses), something [37] glossed over.
Weaknesses
1. For SAT filters to be practical, larger instances need to be solved. This paper only considers instances with 50 variables, allowing for filters with ~400 elements. Practical filters should be able to have millions of elements, something not discussed, and only barely hinted at in the conclusions. I would like to _also_ see results with more variables, at least to get a sense of scale.
I do appreciate the appendix results for k=3 and k=5
2. The results of [37] should be addressed more fully. Use the introduction (or a background section) to explain how you differ. Instead, this was minimally discussed in the Conclusions section.
Report
The paper was enjoyable to read. The topic is very interesting and addresses the open problem of finding disparate solutions to random k-SAT instances. Though the paper focuses on instances coming from the domain of SAT filters, the results are relevant even removed from the domain. See Moshe Vardi's recent work on methods for finding uniform solutions to SAT instances --- http://www.cs.rice.edu/~vardi/papers/cav16tk.pdf
Requested changes
1. Abstract --- walkSAT should be WalkSAT
2. define SAT
3. "promising type of filters" -> "promising type of filter"
4. Introduction --- boolean should always be Boolean
5. "boolean satisfiability problems" -> "Boolean satisfiability problem"
6. "find solutions, which are" -> "find solutions which are"
7. Satisfiability filters --- "It can be queried..." What is "It"?
8. "the appearing variables, is" -> "the appearing variables is"
9. "ki in Equation ??" What is "??"
10. "the solvability of a random..." Solvability is not the right word. The right word is Satisfiability.
11. "is chosen, which map" -> "is chosen which map"
12. Another "Equation ??"
13. Another "Equation ??"
14. "to a clause that is fulfilled" Please stop using a thesaurus to find other ways to say Satisfied. Just say satisfied.
15. "and a FPR p, is defined [2]" The correct reference is [10], not [2].
16. Results and Discussion --- "see section II B" -> Section should be uppercase
Daniel Herr on 2017-01-26 [id 91]
Dear Editor,
We are grateful for the hard work the referees have spent on improving our manuscript. We will go through their suggestions and detail our changes. Furthermore, we changed the layout of the manuscript to the SciPost style guide.
We changed all typos, broken references (mainly to equations) and grammar issues mentioned by the referees. In the following, we want to address the remaining comments.
Referee Comment: 1. For SAT filters to be practical, larger instances need to be solved. This paper only considers instances with 50 variables, allowing for filters with ~400 elements. Practical filters should be able to have millions of elements, something not discussed, and only barely hinted at in the conclusions. I would like to also see results with more variables, at least to get a sense of scale.
Response: This is a valid concern, however we want to test the behavior of QA for SAT problems. The fact that this algorithm finds less separate solutions compared to SA should be independent of system size. Furthermore, the simulations for quantum annealing are very demanding. We already went to the largest possible size that could be run on our supercomputer. The classical annealing code could have been run much faster but the main idea of the paper is to compare the two algorithms. Thus, our quantum annealing simulation restricts the size of problems.
Referee Comment: 2. The results of [37] should be addressed more fully. Use the introduction (or a background section) to explain how you differ. Instead, this was minimally discussed in the Conclusions section.
Response: We agree and added a section to the introduction: “ The D-Wave quantum annealer has already been used for the creation of SAT filters. However, the chimera-graph of the D-Wave device does not allow arbitrary connections and thus restricts the types of problems, that can be implemented without embedding. This paper investigates the question whether QA on an arbitrary connected and completely coherent device, can be advantageous for the creation of SAT filters. ”
Referee Comments: 4 - What are the typical values of k and m. (Generally, a little more grounding of SAT fliters would be helpful to a physics audience.) 5 - Clarify what is meant by "theoretically" in the sentence "Weaver et al. showed that SAT filters can theoretically achieve the limit ..." Does this mean a non-constructive proof was given but no actual construction is known, or known constructions are too computationally expensive, or something else entirely?
Response: This sentence in comment 5 was not clearly formulated. Changing this an example for the value of $k$ is now given: “ Weaver et al.~showed that SAT filters can approach the information theoretical limit ($\mathcal{E}=1$) by increasing the size of its clauses. This also increases the complexity of the problem, but it was shown that even for relatively small values of $k=3$ better performance compared to the constant efficiency of Bloom filters ($\mathcal{E}\leq \ln(2)$) can be achieved. ” The typical value of $m$ is linked to the number of elements in the set $Y$ from which the filter is created. This has been mentioned several times throughout the paper (see beginning of Section II C or Section II D at the definition of the efficiency)
Referee comment: 6 - At the beginning of section III.C., T_0 and T_1 are mentioned, but these have not been introduced earlier. The reader can guess that they are the initial and final temperature, but this should be spelled out.
Response: This is an inconsistency: the inverse temperature should be used throughout the paper. We replaced $T_1$ and $T_2$ by their inverse temperatures $\beta_0$ and $\beta_1$. These were already defined at the beginning of Section III.
Referee comment: 7 - Expand on the possible ways forward, as suggested by some of the comments in the report.
Response: From the well documented report we made changes throughout the paper ensuring that we do not overstate any claims. We added the limitations of our study already to the introduction:
“Thus, our simulations indicate that QA shows no improvement for the construction of SAT filters. However, it should be noted that nonlinear schedules or non-stoquastic driver Hamiltonians could still improve QA. Such research is left for further work.”
We also rewrote parts of the conclusion and added a new paragraph stating that outlines further possible research:
“The previously mentioned nonlinear schedules and non-stoquastic driver Hamiltonians could improve the performance considerably and should be investigated in further research. This research would not only be important for analog devices such as the D-Wave; but would also be fairly indicative of QA run on a digital quantum computer, where the implementation of more complex driving Hamiltonians should be easier.”
This should address several comments made in the report section of the referee.
Referee comment: - The authors mention a few times that combining QA with SQA and walkSAT might be a good way forward. This suggests that even if QA alone does not improve the construction of SAT filters, it can potentially improve the construction of SAT filters through complementing another algorithm.
Response: Our analysis shows, that a mixing of SQA and other algorithms does not give any improvements. This has been explained at the end of Seciton IV B (relating to Figure 5).
Referee comment: A secondary concern is that this paper has been submitted to SciPost Physics, but contains very little physics content. Sure, simulated annealing is a computer science algorithm inspired by classical physics and SQA is a computer science algorithm inspired by quantum physics, but there is essentially no discussion of physics. This paper appears to this reviewer to be a pure computer science paper, albeit one that will be of interest to the quantum computing community. The authors should consider whether insights from physics can be helpful in illuminating the results. The editorial board should consider whether pure computer science papers are appropriate for this new journal.
Response: The main focus of this paper is the efficacy of Quantum Annealing. Among other papers, this should give insights into which problems are susceptible to speedup, using quantum annealers such as the D-Wave machine. Thus, we are of the opinion that this paper is very relevant for physicists and we see no problem for it to be published in a physics journal.
We want to thank the referees again for all the helpful comments and believe that these improved the manuscript considerably.

---

## Round 3 · Referee Report · Anonymous · 2016-12-7

Strengths
1 - Addresses an problem of interest, with significant practical importance.
2 - Thorough analysis of results obtained.
3 - Solid exposition.
4 - Good description of possible future directions.
Weaknesses
1 - Conclusions are too strongly stated, and could discourage future research that might be of value.
2 - Little physics content.
3 - Needs a little more attention to communicating computer science content to physicists who may not be as familiar with the concepts involved as is assumed.
4 - Some sloppiness that can easily be corrected
Report
This paper compares the performance of three classical algorithms, SQA. SA, and walkSAT, on problems related to the construction of satisfiability filters.
My main concern is that the conclusions are too strong for the study described, and may inhibit further work that could potentially be quite interesting, both by inhibiting researchers in looking further and by inhibiting funders from funding such research. For example, at the end of the second paragraph they say “We conclude that QA does not improve the construction of satisfiability filters.” Such a sweeping statement should not be made unless much stronger results or arguments can be put forward. All such statements should be qualified to make clear the limitations of the study. The reason for such care include
- The authors mention a few times that combining QA with SQA and walkSAT might be a good way forward. This suggests that even if QA alone does not improve the construction of SAT filters, it can potentially improve the construction of SAT filters through complementing another algorithm.
- It is widely recognized that for QA to outperform classical methods, the simplest QA algorithm methods are unlikely to suffice. Interest in non-stoquastic Hamiltonians is high, and nothing in this paper suggests that such approaches should not be investigated for potential use in sat filter construction.
- Only a linear QA schedule was used. Such a schedule corresponds to neither the most physically implementable schedule (for example, D-Wave quantum annealers do not use such a schedule) nor the theoretically optimal schedule (theoretical work suggests flatter curves at the beginning and end of the anneal perform better).
- Only a single schedule was used. Multiple schedules could increase the diversity. Beyond varying the schedule for the driver and problem Hamiltonians, past work has shown the efficacy of adding other terms to the Hamiltonian that are zero at both ends, but affect the dynamics in between.
- The classical SQA was used instead of actual QA. While there is strong theoretical work showing that the SQA can be useful in predicting QA performance, the extent to which that analogy holds is unclear, especially for something as little studied as the diversity of solutions.
- Only 20 instances for each k were considered, and these instances did not come from SAT filter application problems, but were instead randomly generated problems at the satisfiability threshold. It is unclear how representative these instances are of the type of instances that will arise in practice.
- QA has two aspects, one as a hardware prescription, the other as a quantum algorithm. The authors comment in their conclusions that this studying is the best case scenario for QA in that any physical QA device would have connectivity restrictions. That is true for near-term quantum annealers, but the QA algorithm is of independent interest – if an idealized version is effective, it can be digitized and run on a universal quantum computer thereby supporting arbitrary connectivity. Thus, quantum annealing as an algorithm could be useful for this problem even if it turns out quantum annealers are not.
The paper is a solid study, but the authors should go through the paper carefully, and modify any statements that overstate their conclusions (there are others than the example given above).
A secondary concern is that this paper has been submitted to SciPost Physics, but contains very little physics content. Sure, simulated annealing is a computer science algorithm inspired by classical physics and SQA is a computer science algorithm inspired by quantum physics, but there is essentially no discussion of physics. This paper appears to this reviewer to be a pure computer science paper, albeit one that will be of interest to the quantum computing community. The authors should consider whether insights from physics can be helpful in illuminating the results. The editorial board should consider whether pure computer science papers are appropriate for this new journal.
Requested changes
1 - Used dashed lines, etc., in the figures so they are more easily readable when the paper is printed in b/w/
2 - Fix broken references, currently appearing as ??, throughout the paper, particularly for Equations.
3 - Sentence near bottom of right hand column of p.2 should read "If none of the stored solutions satisfy the newly ..."
4 - What are the typical values of k and m. (Generally, a little more grounding of SAT fliters would be helpful to a physics audience.)
5 - Clarify what is meant by "theoretically" in the sentence "Weaver et al. showed that SAT filters can theoretically achieve the limit ..." Does this mean a non-constructive proof was given but no actual construction is known, or known constructions are too computationally expensive, or something else entirely?
6 - At the beginning of section III.C., T_0 and T_1 are mentioned, but these have not been introduced earlier. The reader can guess that they are the initial and final temperature, but this should be spelled out.
It is inaccurate to say that the number of solutions found has saturated by 2000 runs in Fig. 1. The slope is still quite positive at the endpoint.
7 - Expand on the possible ways forward, as suggested by some of the comments in the report.

---

## Round 5 · Referee Report · Anonymous (Referee 2) · 2017-2-26

Strengths

See original review.

Weaknesses

See original review.

Report

See original review.

Requested changes

Paper should mention typical size of filter, number of variables used in practice, as requested by both referees and still not in the paper. They are submitting it to a venue in which most of the readers will not be familiar with SAT filters, so providing this information, and commenting on how far these experiments are from realistic values is important to provide context. In general, more recognition that the audience is a physics audience would be appreciated.

---

## Round 5 · List of Changes

Dear Editor,

We are grateful for the hard work the referees have spent on improving our manuscript. We will go through their suggestions and detail our changes. Furthermore, we changed the layout of the manuscript to the SciPost style guide.

We changed all typos, broken references (mainly to equations) and grammar issues mentioned by the referees. In the following, we want to address the remaining comments.

Referee Comment: 1. For SAT filters to be practical, larger instances need to be solved. This paper only considers instances with 50 variables, allowing for filters with ~400 elements. Practical filters should be able to have millions of elements, something not discussed, and only barely hinted at in the conclusions. I would like to also see results with more variables, at least to get a sense of scale.

Response: This is a valid concern, however we want to test the behavior of QA for SAT problems. The fact that this algorithm finds less separate solutions compared to SA should be independent of system size. Furthermore, the simulations for quantum annealing are very demanding. We already went to the largest possible size that could be run on our supercomputer. The classical annealing code could have been run much faster but the main idea of the paper is to compare the two algorithms. Thus, our quantum annealing simulation restricts the size of problems.

Referee Comment: 2. The results of [37] should be addressed more fully. Use the introduction (or a background section) to explain how you differ. Instead, this was minimally discussed in the Conclusions section.

Response: We agree and added a section to the introduction: “ The D-Wave quantum annealer has already been used for the creation of SAT filters. However, the chimera-graph of the D-Wave device does not allow arbitrary connections and thus restricts the types of problems, that can be implemented without embedding. This paper investigates the question whether QA on an arbitrary connected and completely coherent device, can be advantageous for the creation of SAT filters. ”

Referee Comments: 4 - What are the typical values of k and m. (Generally, a little more grounding of SAT fliters would be helpful to a physics audience.) 5 - Clarify what is meant by "theoretically" in the sentence "Weaver et al. showed that SAT filters can theoretically achieve the limit ..." Does this mean a non-constructive proof was given but no actual construction is known, or known constructions are too computationally expensive, or something else entirely?

Response: This sentence in comment 5 was not clearly formulated. Changing this an example for the value of $k$ is now given: “ Weaver et al.~showed that SAT filters can approach the information theoretical limit ($\mathcal{E}=1$) by increasing the size of its clauses. This also increases the complexity of the problem, but it was shown that even for relatively small values of $k=3$ better performance compared to the constant efficiency of Bloom filters ($\mathcal{E}\leq \ln(2)$) can be achieved. ” The typical value of $m$ is linked to the number of elements in the set $Y$ from which the filter is created. This has been mentioned several times throughout the paper (see beginning of Section II C or Section II D at the definition of the efficiency)

Referee comment: 6 - At the beginning of section III.C., T_0 and T_1 are mentioned, but these have not been introduced earlier. The reader can guess that they are the initial and final temperature, but this should be spelled out.

Response: This is an inconsistency: the inverse temperature should be used throughout the paper. We replaced $T_1$ and $T_2$ by their inverse temperatures $\beta_0$ and $\beta_1$. These were already defined at the beginning of Section III.

Referee comment: 7 - Expand on the possible ways forward, as suggested by some of the comments in the report.

Response: From the well documented report we made changes throughout the paper ensuring that we do not overstate any claims. We added the limitations of our study already to the introduction:

“Thus, our simulations indicate that QA shows no improvement for the construction of SAT filters. However, it should be noted that nonlinear schedules or non-stoquastic driver Hamiltonians could still improve QA. Such research is left for further work.”

We also rewrote parts of the conclusion and added a new paragraph stating that outlines further possible research:

“The previously mentioned nonlinear schedules and non-stoquastic driver Hamiltonians could improve the performance considerably and should be investigated in further research. This research would not only be important for analog devices such as the D-Wave; but would also be fairly indicative of QA run on a digital quantum computer, where the implementation of more complex driving Hamiltonians should be easier.”

This should address several comments made in the report section of the referee.

Referee comment: - The authors mention a few times that combining QA with SQA and walkSAT might be a good way forward. This suggests that even if QA alone does not improve the construction of SAT filters, it can potentially improve the construction of SAT filters through complementing another algorithm.

Response: Our analysis shows, that a mixing of SQA and other algorithms does not give any improvements. This has been explained at the end of Seciton IV B (relating to Figure 5).

Referee comment: A secondary concern is that this paper has been submitted to SciPost Physics, but contains very little physics content. Sure, simulated annealing is a computer science algorithm inspired by classical physics and SQA is a computer science algorithm inspired by quantum physics, but there is essentially no discussion of physics. This paper appears to this reviewer to be a pure computer science paper, albeit one that will be of interest to the quantum computing community. The authors should consider whether insights from physics can be helpful in illuminating the results. The editorial board should consider whether pure computer science papers are appropriate for this new journal.

Response: The main focus of this paper is the efficacy of Quantum Annealing. Among other papers, this should give insights into which problems are susceptible to speedup, using quantum annealers such as the D-Wave machine. Thus, we are of the opinion that this paper is very relevant for physicists and we see no problem for it to be published in a physics journal.

We want to thank the referees again for all the helpful comments and believe that these improved the manuscript considerably.

---

## Round 6 · List of Changes

Warnings issued while processing user-supplied markup:
- Inconsistency: plain/Markdown and reStructuredText syntaxes are mixed. Markdown will be used.
Add "#coerce:reST" or "#coerce:plain" as the first line of your text to force reStructuredText or no markup.
You may also contact the helpdesk if the formatting is incorrect and you are unable to edit your text.
Dear Editor,
as suggested by the referee we added a paragraph which illustrates what values (FPR, size of the set, storage requirements) a typical SAT-filter has. We explicitly mention the size of the SAT-problem and the size (in bits) that need to be stored for the solutions of these problems. The paragraph can be found towards the end of section 2.4 Important Quantities:
""" Since filters are mainly used in big data contexts, their size can be quite large, depending on the size of the set $\left|Y\right|$ and the required FPR. To illustrate this, we give an example for a SAT filter encoding a set $Y$ with $\left|Y\right|=2^{16}$ elements: The $2^{16}$ elements result in $m=2^{16}$ clauses. If the required FPR is $p=\frac{1}{4}$ a 4-SAT filter needs to store $s\approx22$ (independent) solutions~\cite{weaver2014}. If the targeted efficiency is given by $\mathcal{E}=0.75$ each solution involves $n \approx 8136$ literals. The filter therefore stores $\approx 22 \cdot 8136$ bits. A 5-SAT filter achieving the same efficiency and FPR needs to store $s\approx 44$ (independent) solutions with $n \approx 4002$ literals. It should be noted that a physical implementation needs more spins as there are literals, because embedding is needed to accommodate for the restricted connectivity of physical devices. Thus, an implementation with current hardware is not yet possible, but the rapid development in this field might lead to physical implementations, soon. """
We again like to thank the referees for their work on our manuscript.
Best wishes, Daniel Herr

---

## Editorial Decision

published